# CBCT Images to an STL Model: Exploring the “Critical Factors” to Binarization Thresholds in STL Data Creation

**DOI:** 10.3390/diagnostics13050921

**Published:** 2023-03-01

**Authors:** Takashi Kamio, Taisuke Kawai

**Affiliations:** Department of Oral and Maxillofacial Radiology, School of Life Dentistry at Tokyo, The Nippon Dental University, 1-9-20 Fujimi, Chiyoda-ku, Tokyo 102-8159, Japan

**Keywords:** cone beam computed tomography, imaging, radiology, oral and maxillofacial surgery, dentistry, DICOM, 3D printing, computer-aided design, STL data

## Abstract

In-house fabrication of three-dimensional (3D) models for medical use has become easier in recent years. Cone beam computed tomography (CBCT) images are increasingly used as source data for fabricating osseous 3D models. The creation of a 3D CAD model begins with the segmentation of hard and soft tissues of the DICOM images and the creation of an STL model; however, it can be difficult to determine the binarization threshold in CBCT images. In this study, how the different CBCT scanning and imaging conditions of two different CBCT scanners affect the determination of the binarization threshold was evaluated. The key to efficient STL creation through voxel intensity distribution analysis was then explored. It was found that determination of the binarization threshold is easy for image datasets with a large number of voxels, sharp peak shapes, and narrow intensity distributions. Although the intensity distribution of voxels varied greatly among the image datasets, it was difficult to find correlations between different X-ray tube currents or image reconstruction filters that explained the differences. The objective observation of voxel intensity distribution may contribute to the determination of the binarization threshold for 3D model creation.

## 1. Introduction

Recently, patient-specific three-dimensional (3D) models have been widely used in clinical practice. In particular, osseous 3D models are used to simulate surgical procedures on delicate anatomical structures during maxillofacial surgery, craniofacial surgery, and otolaryngology, and they are also used for surgical training and medical education. These 3D models allow users to actually manipulate the 3D model and view it from any angle. They can also be simulated with actual surgical instruments. Such an understanding of the three-dimensional anatomy of the surgical site contributes to a better understanding on the part of the surgeon, the surgical team, and medical students. This, in turn, is expected to lead to more predictable surgeries and shorter operating times. Its use as a preoperative educational tool for patients also contributes to better communication [1,2,3,4].

The generalization of 3D printing technologies has facilitated the fabrication of such 3D models from multi-detector row computed tomography (MDCT) images using desktop 3D printers; i.e., “one-stop 3D printing” [5].

Over the last two decades, cone beam computed tomography (CBCT), which has a smaller field of view (FOV) than MDCT and a higher spatial resolution, has become remarkably widespread, especially in the field of oral and maxillofacial surgery [6]. The design of osseous 3D models primarily requires 3D computer-aided design (CAD) data represented in stereolithography (STL) format. Although these data are used to create STL models (3D CAD models), the design of a more precise 3D model is expected to use image datasets from the CBCT scanner as the source of data.

The first step, binarization of the Digital Imaging and Communications in Medicine (DICOM) image dataset, is the most important operation in the sequence of fabricating 3D models. On the basis of numerous 3D model fabrication examples and our previous studies, unlike MDCT image datasets, it is sometimes difficult to determine the value to set the binarization threshold of an STL model using CBCT image datasets [7]. Several factors make the binarization of CBCT image datasets difficult. The biggest problem is that the threshold for binarization varies from one CBCT image dataset to another, and there is no specific value, such as a “CT value”, for an MDCT image. The quality of image datasets exported as DICOM files varies from one CBCT unit to another, which is a major factor affecting the determination of the threshold value. These are also the problems that plague 3D CAD operators in determining the appropriate binarization threshold for creating STL models.

To enable anyone to create 3D models that satisfy clinical demands, it is necessary to establish a method to create highly precise 3D models from CBCT images in a reasonable procedure. It is also necessary to understand the characteristics of image datasets and what factors make it easy or difficult to determine the binarization threshold for STL model creation. Therefore, to understand the factors that affect the determination of the binarization threshold, this study aimed to evaluate the quality of STL models created from image datasets acquired with a CBCT scanner and to explore the factors relating to successful STL model creation from the perspective of 3D image engineering and 3D image processing.

## 2. Materials and Methods

### 2.1. Definition 

In this study, DICOM image datasets exported to the STL file format (or data after segmentation) are called “STL data”; 3D surface CAD models (virtual 3D models) created from STL data are called “STL models”. The model fabricated with a 3D printer from the STL data is referred to as a “3D model”. The units for the intensity (brightness) values of the voxels are expressed in gray values (GV), according to a previous report by Katsumata et al. [8].

### 2.2. CBCT Scanning and Imaging to STL Model Creation

A dry human skull was used as a specimen. The skull was placed in a 20 cm cubic acrylic case filled with water and scanned. A cranial specimen with bilateral missing maxillary molars, without crown restorations, was used for maxillary alveolar and periapical maxillary sinus surgery (Figure 1). Scanning was performed with two different CBCT units, SOLIO XZ II (Asahi Roentgen Ind., Co. Ltd., Kyoto, Japan) and 3D Accuitomo F17 (J. Morita Mfg. Corp., Kyoto, Japan). After scanning, the image processing software attached to each CBCT unit was used to export the images as a DICOM file. Details of each scanning condition and the image reconstruction filters applied are shown in Table 1.

The impact of differences in the scanning X-ray tube current and image reconstruction filters on the determination of the binarization threshold in STL model creation was evaluated. The binarization threshold for each image was set so that both anterior and posterior nasal spines could be visualized and identified. Volume Extractor 3.0 (VE3, i-Plants Systems, Iwate, Japan), a 3D medical/general purpose image processing software package [9], was used to binarize the image datasets and create STL models. In VE3, the Marching Cube method was used as the isosurface extraction method. VE3 was used only for the conversion of image datasets to STL models, not for image linear interpolation or the noise reduction function. The removal of spatial polygonal noise (independent polygonal data not touching the surface of the STL model) was performed using the “Remove small polygons” command in the polygon data editing software package POLYGONALmeister ver7 (UEL Corp., Tokyo, Japan) [10].

### 2.3. Converting the DICOM Image Dataset to 256 Grayscale and Making a Histogram

ImageJ (version 1.53k, NIH, Bethesda, MD, USA) was used to convert the DICOM image dataset to 256 grayscale, and its histogram was used to evaluate the appearance of the voxel intensity distribution.

### 2.4. STL Model Superimposition, Comparison, and 3D Model Fabrication

To find the differences in the shape of each STL model created from the CBCT image dataset, the STL models were superimposed and the shape error (signed difference between two STL models) was observed. An STL model was created from the MDCT image dataset, which was used as the gold standard for the STL model shape. The MDCT scanner was an Aquilion 64 system (Canon Medical Systems Inc., Tochigi, Japan), with the following scanning conditions: X-ray tube voltage of 120 kV, X-ray tube current of 150 mA, slice thickness of 0.5 mm, FOV of 240 mm, a 512 × 512 matrix, and convolution kernel FC81 (bone sharp). The 3D evaluation software package spGauge 2014.1 (spG, ARMONICOS Co., Ltd., Shizuoka, Japan) was used for the superimposition of the created STL models and their color mapping. In the superimposition, one of the two STL models was moved using spG’s best fit surface-based registration algorithm, and the operation was repeated until the misalignment with the other STL model was as close as possible to 0.00 mm. STL models created from CBCT image datasets were fabricated as 3D models using the same procedure as in our previous report [11].

3D model fabrication was performed with a desktop Fused Deposition Modeling (FDM) 3D printer (Value3D MagiX MF-800, Mutoh Industries, Tokyo, Japan). The 3D printing conditions are as follows: a PolyTerra-1.75 mm PLA filament, a laminating pitch of 0.2 mm, an infill density of 30%, a printing speed of 40 mm/s, and with support structures and rafts.

## 3. Results

### 3.1. Visibility of the Exported Images

Each image in the CBCT image dataset was imported and displayed in VE3, as shown in Figure 2. These are the native images with the default display settings of VE3. The same specimen was scanned; however, different X-ray tube currents and different applied image reconstruction filters resulted in different appearances.

### 3.2. Differences in the Shape of Each STL Model

The STL model created from each CBCT image dataset, with the binarization threshold for STL model creation set to the GV where the anterior and posterior nasal spines can be identified, is shown in Figure 3. For each image dataset, there was a difference in the shape of the STL model and the number of polygons that were mixed in as noise.

### 3.3. Histogram of the Voxel Intensity Distribution

Figure 4A,B show the voxel intensity distribution of each CBCT image dataset, and Figure 4C shows the voxel intensity distribution of the MDCT image dataset used as the gold standard for STL model creation. In both CBCT image datasets, each voxel had a single peak; however, the maximum value and range of the peak varied. Histogram observations showed that, compared with the 3D Accuitomo F17, the SOLIO XZ II had fewer maximum voxels and tended to converge at the center of the voxel intensity value distribution. In the image dataset acquired on the Aquilion 64, the distribution of the voxel intensity values was bimodal, with apex-like peaks and narrow ranges. 

### 3.4. Differences in Images for Each GV

The image visibility also differed significantly depending on the binarization threshold. At lower threshold values, the spatial noise increased. However, a higher threshold resulted in less replicability of thin areas of bone tissue (Figure 5).

### 3.5. Shape Error of the STL Models

The superimposed shape error of the STL models created from the two CBCT and MDCT image datasets, respectively, showed that both the SOLIO XZ II and 3D Accuitomo F17 had more noise in the teeth and surrounding tissue, and the error was large. Additionally, the SOLIO XZ II had a rougher surface than the 3D Accuitomo F17, with a higher percentage of green to yellow or slightly distended areas (Figure 6).

## 4. Discussion

Dental implant surgeries, such as maxillary sinus floor augmentation, and maxillofacial surgery, such as a Le Fort I osteotomy, require delicate handling of the maxillary alveolar bone, maxillary sinus, and sphenoid process regions. As a result, highly precise 3D models are necessary for their simulation. In this study, we explored the optimal binarization threshold for STL model creation to fabricate osseous 3D models that faithfully replicate delicate and complicated anatomical structures.

Two flagship CBCT scanners from two different suppliers were used. One is the SOLIO XZ II, a multifunctional scanner capable of CBCT scanning and panoramic imaging. This scanner is mainly used in the field of dentistry and maxillofacial surgery. The other is the 3D Accuitomo F17 scanner, which is a dedicated scanning unit for CBCT, with a range from a minimum φ40 mm × height of 40 mm to a maximum of φ170 mm to a height of 230 mm, and 11 different FOVs can be selected according to the region of interest. The Accuitomo F17 scanner is primarily used in the fields of oral and maxillofacial surgery, craniofacial surgery, and otolaryngology.

Three steps are required to create a 3D model from an image dataset [6,11]. The first step is to acquire the 3D volumetric data of the object as a DICOM file. The second step is to segment the anatomical structures of the 3D model fabrication object from the surrounding structures and export them to a 3D CAD model in STL file format. While segmentation of hard and soft tissues is relatively easy, the subsequent STL model creation is sometimes difficult for two reasons. Small or thin bones (such as in the maxillary sinus, nasal cavity, or orbital floor) and narrow bone cavities (such as the joint cavity of the temporomandibular joint) are not easily reflected in the STL model. Problems resulting from the scanning characteristics of CBCT/MDCT (e.g., metallic artifacts from dental materials, beam hardening, overshooting, and undershooting) reduce image visibility and make it difficult to determine the threshold for binarization. The final step is to generate data for 3D printing, called G-code, from the created STL model and run the 3D printer. Failure in any of these steps results in poor quality of the 3D model. Therefore, the creation of satisfactory STL data is the most important operation in the sequence of 3D model fabrication [6,12].

CBCT images do not contain the quantitative physical quantities that are in MDCT images, such as “CT values” or “Hounsfield units” [13,14,15]. In other words, because the GV of a CBCT image is not a linearized value, the binarization threshold is also difficult to determine using standard methodologies. Therefore, the shape of the STL model may be highly dependent on the discretion of the 3D CAD operator. The 3D model created from the STL model (A6M), which the authors created by determining the binarization threshold, is shown in Figure 7. Even with the full use of polygon data editing software packages, it was difficult to control the noisy 3D CAD model. The detail of the fabricated 3D model is somewhat less replicable, with numerous irregular structures appearing on the surface.

With our approach, differences in the visibility of each X-ray tube current and image reconstruction filter—that is, differences in the scanning and imaging characteristics of each CBCT unit—could be objectively visualized as differences in the intensity distribution of voxels. This is the same as a visualization of the differences in image contrast in each image dataset. These differences in image datasets can be attributed to differences in the method of image reconstruction performed by image processing software after CBCT image data acquisition; i.e., differences in the image gamma correction applied to each filter. It was easier to determine the binarization threshold of an image dataset with high image contrast that was made in the histogram as a positive curve with more voxels, high peaks, a relatively narrow distribution, and a steep slope. Unsuitable binarization thresholds can result in poor replication of details in the STL model. 3D models from poor-quality STL models can, for example, lead to unintended defects or cavities, thus compromising the reliability of the surgical simulation. This result partly explains why it is difficult to determine the binarization threshold when the peak number of voxels is too low or the voxel intensity distribution range is too narrow or too wide. Although the binarization threshold required for STL data depends on the application of the 3D model and cannot be uniformly specified, a histogram of the voxel intensity distribution of the image dataset would help determine the binarization threshold for creating STL models that satisfy clinical demand.

There are numerous reports on automatic segmentation of CBCT and MDCT images. However, even for MDCT images, for which voxel intensity values can be obtained quantitatively, automatic segmentation is difficult to perform. For CBCT images, however, there are many different scanners available, with varying scanning and imaging characteristics, making segmentation a very difficult challenge [16,17,18,19,20,21,22]. Furthermore, to our knowledge, there are few reports evaluating MDCT images for STL data creation for 3D model fabrication, and there are no reports using the same approach as ours using CBCT images [23,24,25,26]. CBCT scanners are expected to be used as 3D scanning devices for dental casts and dental prosthetics (surgical guides, dentures) [27,28,29]. As the demand for 3D printed models increases, better STL data are needed and their importance will increase. Because the two CBCT units used in this study were not equipped with image reconstruction filters that are specialized for the 3D model, such as the “STL data creation mode”, the binarization threshold for creating STL data had to be determined manually by the 3D CAD operators. Considering the rapid development of 3D technology, a CBCT unit equipped with an “Imaging filter for 3D copy” that can create optimal STL data under optimal scanning and imaging conditions will likely be available soon.

A limitation of this study is that there were only 18 image datasets with each scanning X-ray tube voltage set to 85 kV and three representative image reconstruction filters. Both CBCT units had other scanning modes. In addition to the scanning conditions, such as the X-ray tube voltage and tube current, it is possible to select a larger or smaller FOV according to the object size and image reconstruction filters according to the diagnostic purpose. Another limitation is that there is no standard for determining the binarization threshold for creating STL models from CBCT images. We have used a threshold where the targeted anatomical structures are somewhat visible. However, we recognize that this binarization threshold value was determined by the experience of the authors, who are also 3D CAD/3D printing operators, and it is lacking objectivity. There is currently no methodology to prove this binarization threshold is appropriate; however, that is a topic for future research.

In this study, we evaluated voxel intensity distribution histograms made from DICOM image datasets from two different CBCT scanners to determine why it is difficult to create STL data for a 3D model from CBCT images and how to address this issue. The image appearance varied greatly depending on the scanning and imaging conditions, and the binarization threshold also varied relative to each image.

In conclusion, the results of this study suggest the importance of understanding the scanning and imaging characteristics of each CBCT device in advance to determine the optimal and objective binarization threshold. Our approach, which enables facile observation of the visibility of the voxel intensity distribution and is technically simple, may help find the “critical factors” to binarization threshold determination, which is essential for the fabrication of 3D models for the medical field.

## Figures and Tables

**Figure 1 diagnostics-13-00921-f001:**
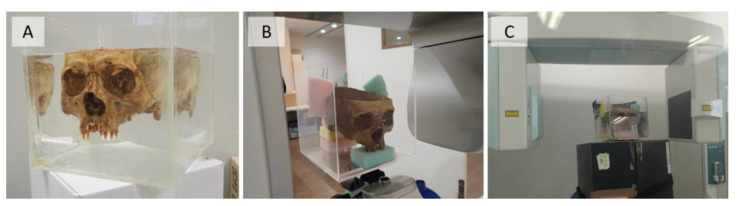
The specimen and CBCT units. (**A**) A specimen in an acrylic case filled with water, and the specimen in the (**B**) SOLIO XZ II (Asahi Roentgen Ind., Co. Ltd., Kyoto, Japan) scanner, and (**C**) 3D Accuitomo F17 (J. Morita Mfg. Corp., Kyoto, Japan) scanner.

**Figure 2 diagnostics-13-00921-f002:**
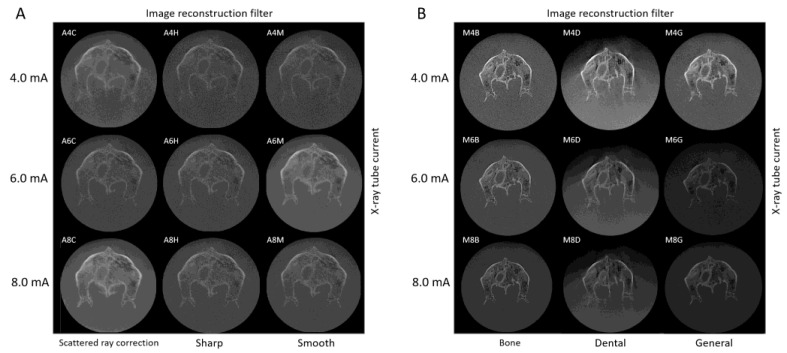
CBCT horizontal cross-sectional images of anterior and posterior nasal spines without window level setting displayed in Volume Extractor 3.0 using the (**A**) SOLIO XZ II scanner and (**B**) 3D Accuitomo F17 scanner.

**Figure 3 diagnostics-13-00921-f003:**
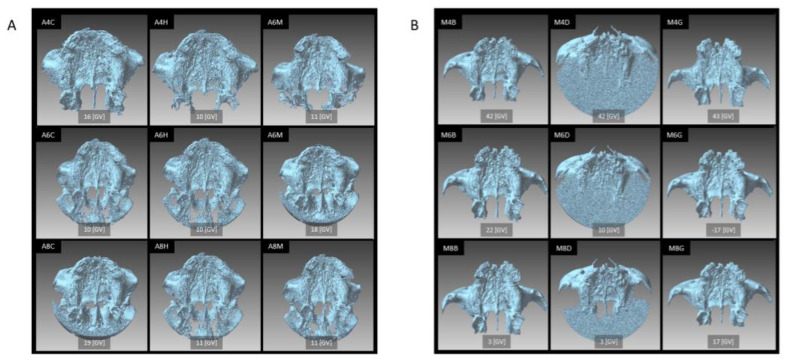
The STL models from the (**A**) SOLIO XZ II scanner and (**B**) 3D Accuitomo F17 scanner were created with the binarization threshold set to an arbitrary value. The abbreviation of each STL model is shown at the left shoulder of each figure, and the binarization threshold is shown at the bottom center of each figure. The GV indicates the percentile after the intensity value of each image dataset was converted to 256 grayscale.

**Figure 4 diagnostics-13-00921-f004:**
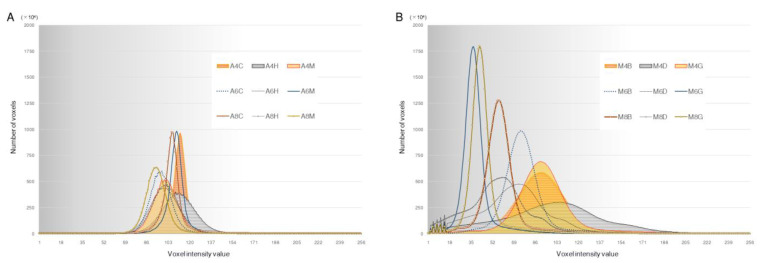
Histograms of voxel intensity for CBCT images from the (**A**) SOLIO XZ II and (**B**) 3D Accuitomo F17 scanners and (**C**) Aquilion 64 MDCT scanner images. The vertical axis represents the number of voxels and the horizontal axis represents the voxel intensity value in 256 grayscale.

**Figure 5 diagnostics-13-00921-f005:**
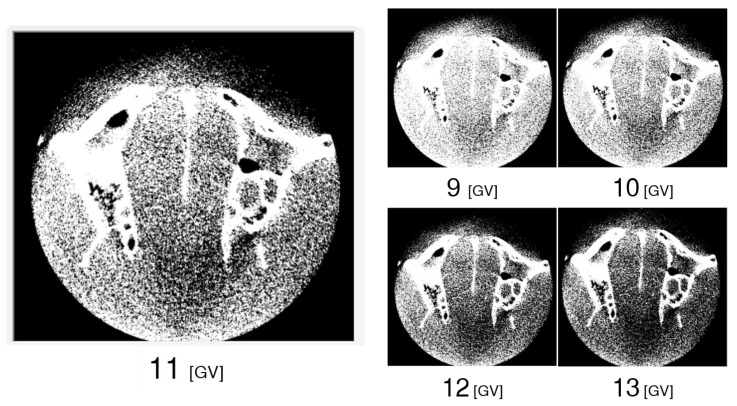
Images at different binarization threshold values (image dataset: A4C). The gray values shown at the bottom of each figure are the native voxel values of the CBCT image (in this case, from 3721 to −2359 [GV]) converted to 256 grayscale.

**Figure 6 diagnostics-13-00921-f006:**
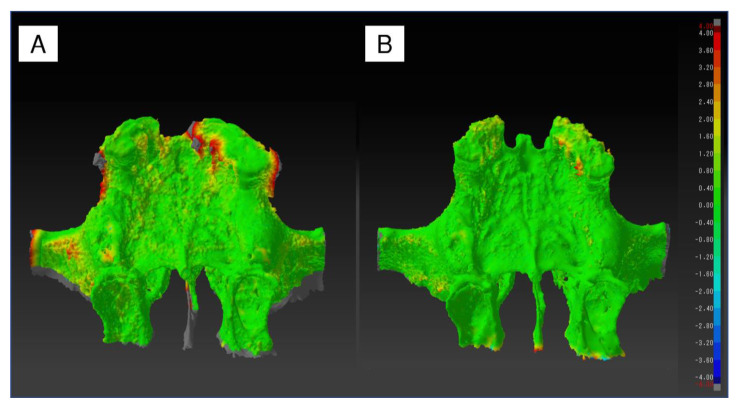
Shape error of two superimposed STL models. (**A**) CBCT STL model (image dataset: A4C) vs. MDCT STL model. (**B**) CBCT STL model (image dataset: M8G) vs. MDCT STL model. The areas where shape errors compared with the MDCT STL model used as the gold standard were observed are in color. Positive errors (expansion) are shown in warm colors and negative errors (contraction) are shown in cool colors.

**Figure 7 diagnostics-13-00921-f007:**
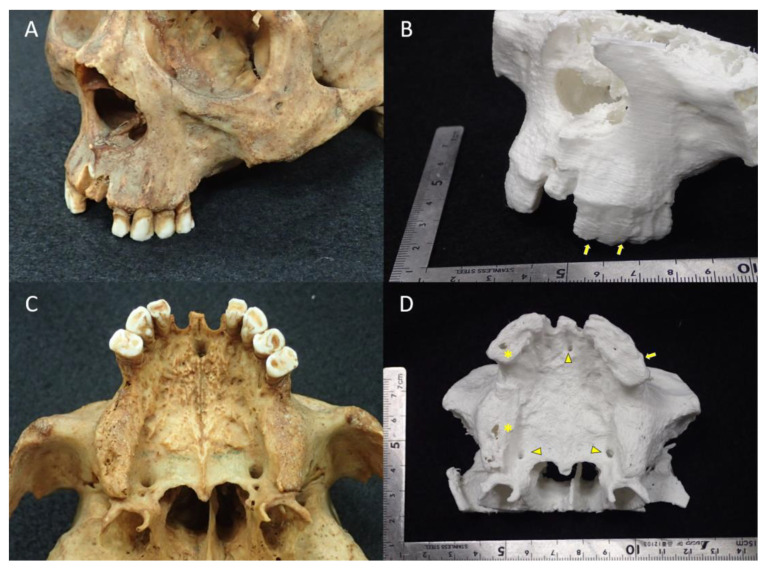
(**A**,**C**) Close-up view of the specimen. (**B**,**D**) Close-up view of the 3D model fabricated from an STL model (image dataset: A4C) with a desktop FDM 3D printer. In fabricating the 3D model, the STL model was smoothed, noise-reduced, and polygon-count-reduced using POLYGONALmeister Ver 7. Observations of the 3D model show that the replication of the teeth and surrounding tissues is poor (arrows). The incisal and greater palatine foramen are also generally narrower (arrowheads). Partial defects on the surface of the 3D model because of 3D printing errors are noted (*).

**Table 1 diagnostics-13-00921-t001:** Overview of the scanning and image reconstruction filters for each CBCT unit and abbreviations for each DICOM image dataset.

CBCT Unit	X-ray Tube Voltage	Scanning Mode Field of View Voxel Size	Image Processing Software (Version)	X-ray Tube Current	Image Reconstruction Filter	Abbreviation
SOLIO XZ II	85 kV	I-MODE φ90 × 91 mm 0.177 mm	NEOPREMIUM2 (NeoExpCalc 1.0.17.0)	4.0 mA	Scattered ray correction	A4C
					Sharp	A4H
					Smooth	A4M
				6.0 mA	Scattered ray correction	A6C
					Sharp	A6H
					Smooth	A6M
				8.0 mA	Scattered ray correction	A8C
					Sharp	A8H
					Smooth	A8M
3D Accuitomo F17	85 kV	D140 × H100 Hi-Fi φ140 × 100 mm 0.200 mm	i-Dixel (3DXAPP 2.3.7.5)	4.0 mA	Bone	M4B
					Dental	M4D
					General	M4G
				6.0 mA	Bone	M6B
					Dental	M6D
					General	M6G
				8.0 mA	Bone	M8B
					Dental	M8D
					General	M8G

## Data Availability

The data presented in this study are available on request from the corresponding author. The data are not publicly available because they belong to an academic institution.

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
