# Peer review of "CBCT Images to an STL Model: Exploring the “Critical Factors” to Binarization Thresholds in STL Data Creation"

_diagnostics, 2023, doi:10.3390/diagnostics13050921_

Round 1

Reviewer 1 Report

Dear authors !

I was given the opportunity to review Your manuscript presenting the results of the accuracy of STL-Data-Sets derived from two different CBCT devices compared with MDCT derived STL-Data-sets and pointing out the difficulty of lack of standardization for highest possible precision/accuracy in 3D reconstruction of Dicom-images for use in CAD/CAM.

The entire study is scientifically sound, very well executed and of high value for future developments and improvements in CAD/CAM-procédures in dentistry and OMF-surgery.

The manuscript is well written in comprehensible English.

Thank You 

Author Response

Thank you for your careful peer review. In order to improve this manuscript, we have added and revised the sections that were pointed out by the reviewers. Corrections and additions that have been made are highlighted in yellow throughout the revised manuscript. They are listed below.

R2-1. Figure 1 should be explained a little more.
A1. In 2.2 CBCT scanning and imaging to STL model creation (P2L73-L74), we added a reason as to why this specimen was used. The revised part is shaded in yellow.

R2-2. [1-4] these papers should be written in detail.
A2. In P1L34-L38, at the beginning of the Introduction, we have added a detailed description of the advantages of using 3D models for the articles cited in this issue [1-4].

R2-3. [6,12]=>[6, 12].
A3. The relevant section of P7L216 has been corrected.

R2-4. [16-22] these papers should be written in detail.
A4. From P8L235-P9L261, we have added an explanation of the current problems pointed out in the articles cited in this study.
While there are many reports on segmentation, there are only a few papers related to STL data generation, and it was actually difficult to find reports on these topics.

R2-5. They need to update the reference section.
A5. A report on making STL models and 3D models of the mandible from CBCT images has been added to reference number 23.

R2-6. They studied on the image processing. 
A6. We are aware of the fact that there are countless reports on image processing for the binarization and segmentation of CT images. We believe that the importance of this article lies in the fact that it aims to "enable anyone to create STL models using a not-so-special (and not expensive) tool" and explains the technical background of image processing that should be considered when doing this. In practice, segmentation can be done without the need for deep knowledge of image processing. And it is possible to create 3D models. However, to create more accurate STL data, we believe it is important to have some knowledge of image processing. In this article, we will explain ( although briefly...) how to do this.

Reviewer 2 Report

See my attached referee report.

Author Response

(The authors gave the same response as above.)
